# A Cloud Enabled Crop Recommendation Platform for Machine Learning-Driven Precision Farming

**DOI:** 10.3390/s22166299

**Published:** 2022-08-22

**Authors:** Navod Neranjan Thilakarathne, Muhammad Saifullah Abu Bakar, Pg Emerolylariffion Abas, Hayati Yassin

**Affiliations:** Faculty of Integrated Technologies, Universiti Brunei Darussalam, Gadong BE1410, Brunei

**Keywords:** smart agriculture, artificial intelligence, machine learning, deep learning, Internet of Things, IoT, cop recommendation, cloud computing

## Abstract

Modern agriculture incorporated a portfolio of technologies to meet the current demand for agricultural food production, in terms of both quality and quantity. In this technology-driven farming era, this portfolio of technologies has aided farmers to overcome many of the challenges associated with their farming activities by enabling precise and timely decision making on the basis of data that are observed and subsequently converged. In this regard, Artificial Intelligence (AI) holds a key place, whereby it can assist key stakeholders in making precise decisions regarding the conditions on their farms. Machine Learning (ML), which is a branch of AI, enables systems to learn and improve from their experience without explicitly being programmed, by imitating intelligent behavior in solving tasks in a manner that requires low computational power. For the time being, ML is involved in a variety of aspects of farming, assisting ranchers in making smarter decisions on the basis of the observed data. In this study, we provide an overview of AI-driven precision farming/agriculture with related work and then propose a novel cloud-based ML-powered crop recommendation platform to assist farmers in deciding which crops need to be harvested based on a variety of known parameters. Moreover, in this paper, we compare five predictive ML algorithms—K-Nearest Neighbors (KNN), Decision Tree (DT), Random Forest (RF), Extreme Gradient Boosting (XGBoost) and Support Vector Machine (SVM)—to identify the best-performing ML algorithm on which to build our recommendation platform as a cloud-based service with the intention of offering precision farming solutions that are free and open source, as will lead to the growth and adoption of precision farming solutions in the long run.

## 1. Introduction

Over the years, by means of human collaboration and with a human touch, traditional agriculture has been transformed into a whole new form that offers greater advantages for the survival of humans. Agriculture, which is considered to be the oldest and primary industry in the world, provides foods and livestock that are needed to feed the world’s population of billions [1]. Today, other than technology services and crude oil, the Gross Domestic Products (GDPs) of many countries globally depend on the production of agricultural goods, highlighting its necessity as a key industry. Over the years, with the adoption of machinery and technology, most of the manual labor work in agriculture has been replaced to a great extent, thus improving overall quality and efficiency, and encouraging more people to participate in agriculture for their livelihood [2].

With the growth in urbanization, there will be a dramatic decrease in arable land in the coming years, raising doubts as to whether it will be possible to meet the demand for agricultural food production. On the other hand, according to the most recent studies, it is evident that current agricultural food production needs to be increased by more than 70% by the year 2050 in order to feed the growing global population [1,2,3]. Thus, owing to various reasons, such as the decrease in arable land, the requirement of manual labor, and the increasing capital costs, meeting the demand for agricultural food production is increasingly becoming a significant challenge [3]. This results in a perfect gap for academia, as well as research and development organizations, to find novel solutions that will make it possible to increase the amount of quality harvest while requiring fewer resources, so that these challenges can be overcome in the long run.

To date, with the aim of increasing the quality and the amount of the harvest, various enabling technologies are being used that are powered by Information and Communication Technologies (ICT), including the Internet of Things (IoT), AI, cloud computing, edge computing, fog computing, and 5G communication technologies. The adoption of these technologies has become a booming trend in recent years owing to the benefits they provide to farmers. Furthermore, this fruitful collation of technologies has paved the way for the development of smart agriculture, which describes the use of smarter technologies for agriculture, with the aim of making farming tasks more efficient [3,4,5].

In the context of modern agriculture, the lack of proper planning, improper harvesting, irregular irrigation, and unpredictable weather conditions such as floods and droughts are the major concerns preventing farmers from meeting their goals, and these can be ameliorated by using AI to assist farmers in making timely decisions [6,7]. At times, poor outcomes in farming and broken expectations can lead to stress and discomfort for ranchers, and may even lead to suicidal thoughts and eventually loss of lives, as is a reality in most developing countries, including Sri Lanka, India, and Bangladesh [7,8,9,10]. Nevertheless, it can also lead to social chaos and affect the economy of countries, as was clearly proved by the economic and food crisis that occurred in Sri Lanka in 2022, with the decision taken to ban the import of all chemical fertilizers into the country as a government policy [10,11]. On the whole, agricultural food production in recent years has faced immense challenges, owing to supply chain and logistics issues arising during the COVID-19 global pandemic, a deadly virus outbreak that is still prevalent [11]. Moreover, the current conflict in the Black Sea region and the supply chain disruptions in the agricultural commodities market have also increased the risk of food insecurity [10,11].

According to the United Nations Food and Agriculture Organization (FAO), nearly 33% of all food produced for human consumption is wasted every year owing to various factors [9,10,11]. These losses can mainly be attributed to the choice of unsuitable crops, lack of proper planning, changes in climate, weeds, pests, changes in government policy, etc. Nevertheless, in recent years, there have been drastic climatic changes occurring owing to global warming [12,13,14,15]. Among all these factors, the selection of unsuitable crops has a great effect on the expectations of farmers, as it burns through the entirety of the resources (such as the cost of seeds, fertilizers, etc.) [6,7,8,9,10,15,16,17,18] that have been spent on harvesting, leading to even more disastrous consequences. Hence, it is indeed essential to prioritize which crop should be harvested before carrying out land preparation, which can be highly challenging to guess on the basis solely of the knowledge gained through traditional farming practices. With the advancement of technology, as mentioned above, novel technologies have been applied in farming to improve the overall health condition of crops and aid farmers throughout the farming process, from land preparation to the preparation of the harvest for market. This portfolio of technologies is commonly known as precision farming or precision agriculture, and is mainly governed by three key technologies: IoT, AI, and agriculture robotics. AI, being a remarkable and revolutionizing technology that mimics typical human thinking processes, aids in making timely and precise decisions that will result in better yield and a higher-quality harvest. Thus, motivated by the manner in which ML, which is a key founding technology of AI, can reshape traditional farming, in this study we aim to present a cloud-hosted ML-powered crop recommendation platform for farmers, so that farmers can have a better sense before commencement of harvesting regarding which crop to harvest, thereby reducing the overall harvest wastage and resulting in better yield and a higher-quality harvest in a timely manner. Thus, motivated by the manner in which ML can assist in precision farming and how it can assist in making timely decisions regarding farming, below we present the key motivational factors behind carrying out the research.
Even though there have been recent studies on various ML applications in smart agriculture, these have only provided a theoretical overview of the application and only focus on experimental evaluations, not implementations.Most of the work carried out has been focused on and limited to publications, and have not addressed the aspect of how these technologies can be offered to farmers for free and as open-source solutions.

### 1.1. Contributions of the Study

As outlined above, with to the aim of aiding farmers in making precise and timely decisions with respect to their farming process, the key contributions of this study can be enumerated as follows.
After the introduction, a quick overview of precision farming is offered, as it is the primary focus of this study.The role of AI in precision farming is addressed, with a special emphasis on the application of ML.To validate our work and differentiate it from the work of others, we provide a brief comparison of recent related work, highlighting the key contributions and the main application areas related to precision farming.We designed a cloud-based ML-driven crop recommendation platform and provide a discussion on how to offer such technologies to farmers for free, with the intention of encouraging researchers who are engaged in this area towards the invention of novel solutions for revolutionizing agriculture.

### 1.2. Outline of the Study

The paper is organized as follows. Following the introduction, we provide a brief overview of precision farming in Section 2, while also providing a brief overview of AI in precision farming, mainly highlighting the ML aspects of AI. Further, in Section 3, a brief literature review is provided, highlighting the latest research in the field, and differentiating our work from theirs. Next, in Section 4, our research methodology is highlighted on the basis of an experimental evaluation of our research, followed by a discussion. Finally, the paper concludes with the conclusions derived through our research work.

## 2. Precision Farming

Precision farming, otherwise known as the precision agriculture, is the next big revolution in agriculture. It aims to bring real-time information on farms and livestock to the farmers as required, allowing them to make precise and timely decisions, resulting in higher harvest and less wastage of scarce resources [12,19,20,21,22]. Predominantly, precision farming is a collation of three main technologies: Artificial Intelligence (AI), agriculture robotics, and Internet of Things (IoTs) [13], as depicted in Figure 1.

In precision farming, a variety of IoT sensors are used to gather various environmental parameters related to farms and livestock, including soil fertilizer level, water requirements, soil nutrient level, and health of animals [1,2,3,4,6,7,8,9,10,12,13,14,15]. The data collected by the various sensors at the end nodes are sent to the cloud or remote servers through wired or wireless communication media. At the cloud or server side, various data analytic methods are utilized to infer useful meanings and interpretations from the data, which are then used to make precise and accurate decisions. Accordingly, the system may order agriculture robots to execute certain tasks in a timely manner. For better understanding, Figure 2 depicts the overall steps in precision farming from the gathering of data from IoT sensors and execution of tasks by agricultural robots, based on understandings of the analyzed data. Additionally, these data, when analyzed and refined, may also offer valuable insights to farmers, including with regard to the condition of crops, plant and animal diseases, and weather conditions, as well as forecasting future conditions and predicting the crop yield, with the aim of maximizing the overall efficiency of the farm [22,23,24,25,26].

At the present time, precision farming solutions are heavily used to increase productivity and maximize crop yield, and the entire crop cycle can benefit from the accurate deployment of precision farming applications. According to Libelium [9], which designs key IoT technological solutions for smart agriculture and other related IoT markets, the total market value for precision agriculture solutions has now almost doubled with respect to that in 2016. There are a lot of startup companies that have been established in recent years that offer various commercial precision agricultural services, both hardware and software solutions, especially in India, Australia, and New Zealand [11]. However, despite the availability of many precision agricultural solutions, most farmers are still reluctant to move forward with the technology, which hinders the digitalization progress of farming for the betterment of humankind.

In precision farming, autonomous robots may perform a variety of tasks, and it is evident that they can replace human laborers when performing most agricultural tasks, such as land preparation, seeding, planting, and harvesting [9]. The autonomous devices commonly used in precision agriculture can be mainly divided into two categories: fully autonomous devices and semi-autonomous devices, such as Unmanned Aerial vehicles (UAVs) and agriculture robots used for detecting plant diseases and weeds [10,11,12,13]. UAVs hold a key place in precision agriculture, as they can gather a vast amount of data on a large-scale farm within a very short period of time, making them an ideal solution for large-scale framing. Moreover, aerial images taken from satellites can also be used in precision farming for identifying suitable land, plant diseases, predicting weather conditions, and remote sensing applications [4,5,10,11,12,13].

Apart from crop condition monitoring and management, livestock management is another important aspect of precision farming, where it can help in monitoring overall health condition and real-time location of animals [13], and improve the productivity, welfare, and reproductive behavior of animals throughout their life cycle. Various intelligent sensors implanted internally and externally on animals and real-time cameras can assist in making smarter decisions regarding underlying conditions and act accordingly in a timely fashion [26,27,28]. 

Despite the slow adoption of precision farming solutions, the wide use of precision farming solutions around the world can be mainly attributed to the power of AI, which is backed by both ML and Deep Learning (DL), the two main pillars of AI. Nonetheless, the availability of high-speed Internet, low-budget sensors, and efficient computational devices has aided the wide dissemination of precision farming solutions at the present time [28,29,30,31,32]. Having provided a brief overview of precision farming, in the next subsection, we will briefly discuss the use of AI in precision farming.

### AI in Precision Farming

AI is a major technology of the 21st century, and it is used by most industries, including in agriculture, surveillance, military, smart city, and healthcare, to make precise decisions on the basis of the underlying conditions and to act accordingly. In general, AI provides computational intelligence such that the machines can learn, understand, and respond according to varying situations. AI can be further categorized into ML, DL, natural language processing (NLP), computer vision, fuzzy logic, expert systems, and swarm intelligence (SI), which are key subfields of AI [12]. As mentioned above, AI is currently applied in a variety of aspects of human life, and even with smart mobile devices like Apple, Samsung, and Microsoft, serving as human, friendly virtual assistants [4,5,6,7,8]. At the current time, according to the latest studies, it is evident that, at the current growth rate of technology, AI is going to change the world more than anything in the history of mankind [9,12,32,33,34,35]. Being the key pillar of precision farming, AI is currently involved in many precision farming applications, allowing farmers to act in a timely manner. On a typical farm, IoT sensors and UAVs produce millions of data points in a single day, accumulating a large volume of data, also referred to as big data [8,9]. In most cases, this big data will be transferred into the cloud, and AI will be used to infer the meaning of this data [35,36,37,38,39].

In precision farming, the data captured from IoT sensors deployed in the field are used to predict crop yield, other related natural weather conditions, and the occurrence of disastrous situations with the help of AI, which will eventually help in meeting the current demand for agricultural food production in the long run [40,41,42]. Hence, it is deemed essential to embrace these precision farming solutions as much as possible. As the main focus of this paper is to present how ML is involved in precision farming by developing an ML-powered crop recommendation platform that can be used by farmers to determine what crop should be harvested on the basis of the known environmental parameters, next, we intend to focus more on the application sides of ML in precision farming, in order to give a better holistic view of ML. In general, ML allows learning without needing to be explicitly programmed, and mimics human problem-solving ability. ML acts as an important decision-making tool in precision farming, and can be applied throughout the entire growing and harvesting cycle. On the whole, this begins with crop prediction, soil preparation and selection, water requirement prediction, crop yield prediction, and finally agricultural robots pick up the harvest by determining the ripeness and the quality of fruits through computer vision techniques [9,39,40,41,42]. 

Normally the process of ML can be broken down into three parts: data loading and preprocessing, model building, and generalization, as shown in Figure 3. The data are loaded in the form of a raw dataset; secondly, the data are preprocessed; and thirdly, the predictive ML model is built using suitable ML algorithms. Finally, the generalization involves predicting the output for inputs on which the ML algorithms have never trained before.

As of now, owing to the availability of powerful innovative algorithms, big data, and fast Internet connections, ML applications have been widely used for solving a variety of problems that humans often fail or need a lot of time to solve. On the other hand, DL, which is a branch of ML, trained on much larger datasets or with higher volumes of big data, can also be used to make intelligent decisions, the same as ML. ML can be further broken down into three categories: supervised learning, unsupervised learning, and reinforcement learning [1,2,3,4,6,7,8,9,10,12,13], as depicted in Figure 4.

Supervised learning is the process of learning or training with well-known labeled data to classify outcomes or predict outcomes accurately [1,2,3,4]. As input, data is fed into the model, and the weights are adjusted until the model is properly fitted, which happens during the cross-validation phase. Further, the supervised learning algorithms used for predicting the categorical values are known as classification algorithms, and the algorithms that are used for predicting the numerical value are known as regression algorithms. Unsupervised learning algorithms work with unlabeled data, in contrast to supervised learning algorithms, and they are capable of discovering unknown objects by precisely grouping similar objects [1,2,3,4]. On the other hand, the implementation of unsupervised algorithms is quite difficult compared to supervised learning algorithms, as the main objective of unsupervised algorithms is to extract hidden knowledge from the training data. When it comes to reinforcement learning, it is a method based on rewarding desired behaviors and punishing undesired behaviors. A reinforcement learning agent, in general, is capable of perceiving and interpreting its surroundings, taking actions, and learning through trial and error [3,4,5,6,7]. For better understanding commonly used ML algorithms are further described in Table 1, in the following.

Supervised learning, unsupervised learning, and reinforcement learning techniques are, taken together, used heavily in various industries, such as in agriculture, in combination with IoT for data analytics. Furthermore, in precision farming, wireless sensor networks (WSN) and IoT are widely combined with ML to quantify and understand the big data generated from the sensing devices. As per the literature, ML applications in precision farming can be mainly apportioned to four key categories—crop management, water management, soil management, and livestock management—which are discussed in detail in the following, and we intend to provide examples for these applications in the next section, summarizing the latest research work.
Applications for crop management

It is imperative that farmers have the information necessary to properly forecast crop output and determine the means by which yield might be increased and how the condition of crops can be managed throughout the entire crop cycle [13]. Temperature, humidity, rainfall pattern, type and quality of the soil, fertilizer, and harvesting pattern are the key driving factors that have a great impact on predicting the condition of the crops and provide insight on how the harvest can be increased [1,2,3,4,10,13]. Nonetheless, during the whole crop life cycle, farmers must pay close attention to the health of crops, since pathogenic fungi, germs, and bacteria get their energy from the crops they grow on, which ultimately affects the harvest as they feed on crops [11,12,13]. Thus, farmers stand to lose a lot more money if the problem is not caught and fixed quickly [10]. When illnesses are eliminated and crops are restored to their former functioning, farmers bear the bulk of the costs in the form of pesticides, which in return have a negative consequence on the surrounding environment [10,11,12,13]. In this regard, with the support offered by IoT and enabling technologies, ML applications provide precise insights on what crop should be planted according to the environmental conditions [1,2,3,4,5], predicting crop diseases and pests [10,11,12,13], predicting the yield and forecasting the forthcoming environmental conditions [1,2,3,4,6,7,8,9,13,14,15,16,17].
Applications for water management

To compensate for rainfall shortages, fresh water is required for irrigation and the delivery of nutrients for plant development, and agricultural activities around the world use over 70% of the available freshwater [2,3,4,5,6]. This emphasizes the significance of the responsible management of water via the use of precision irrigation methods underpinned by ML. Farmers are now dealing with a variety of irrigation issues such as over-irrigation, under-irrigation, water depletion, floods, and so on; and with the adoption of ML and IoT, higher crop yield can still be achieved, while simultaneously reducing the amount of water that is used up in the cultivation process (e.g., adopting ML-powered drip irrigation methods and sprinkler irrigation methods [11,12,13]).
Applications for soil management

The forecast of soil attributes is the first and most significant step in the process of farming, which often influences the selection of seeds and crops, preparation of land and fertilizer, and manure selection [1,2,3,4,6,7,8,9]. As soil characteristics are directly related to the geographical and climatic conditions of the area being utilized, this is an important factor to consider before starting farming on the field. The major components of predicting soil characteristics using ML include forecasting the nutrients in the soil [1,2,3,4], the humidity of the soil surface [11,12,13], and the climatic conditions that will occur throughout the crop’s lifetime [11,12,13].
Applications for livestock management

Livestock production refers to the cultivation of domesticated animals (such as pigs, cattle, sheep, and so on) for the purpose of providing commodities for human consumption such as eggs, milk, and meat. Livestock production and management are dependent on farming aspects of the animals, such as their health, food, nutrition, and behavior [12,13], so that the livestock output can be maximized, and farmers can gain a higher profit. In the current context, IoT, ML, and blockchain technologies are being widely explored to improve livestock sustainability and for analysis of their chewing habits, eating patterns, and movement patterns (e.g., standing, moving, drinking, and feeding habits) [11,12,13], which indicate the amount of stress the animal is experiencing and, in turn, help in predicting the vulnerability of livestock to disease, weight gain, and mortality. According to [12], ML-powered weight forecasting systems are used for the evaluation before slaughter. According to the findings of [5,6,7,8,9,11,12,13], with the support of precision arming solutions powered by ML farmers have the ability to modify their livestock’s diet and living conditions in order to facilitate better growth for the animals in terms of their health, behavior, and weight gain, which will, in turn, improve the economic efficiency of livestock.

## 3. Literature Review on Precision Farming Applications

In recent years, with the adoption of ML in precision farming, several research works have already been conducted in various aspects of agriculture. Thus, in order to give a better overview and to differentiate our work from theirs, in the following we summarize the latest research in a tabular form, in Table 2.

## 4. Materials and Methods

The first step involved in the design of our crop recommendation platform includes preparing our crop recommendation dataset, which we took from Kaggle [22], and was built by augmenting actual rainfall, climate, and fertilizer data available for India [22], and secondly preprocessing the data for further analysis. Altogether, the dataset we employed contained 2200 records and eight features. In the third step, we performed an exploratory data analysis using the underlying data in the dataset, in order to understand the nature of our data. The fourth step involved extracting the best features in order to build our ML models using different classification ML algorithms including KNN, DT, RF, XGBoost, and SVM algorithms. Once the model-building step was completed in the next step, we evaluated the performance metrics of the models, and once the evaluation was complete, we arranged the deployment of our best-performing model as a cloud-based web app on the Google Cloud platform, making up our crop recommendation platform. To achieve a better understanding, all steps involved in the design of our crop recommendation platform, along with the high-level architecture of our proposed platform, are illustrated in Figure 5.

### 4.1. Dataset Preparation and Feature Selection

The features in the chosen crop recommendation dataset [22,23] included soil nitrogen level (N), soil phosphorus level (P), soil potassium level (K), air temperature, air humidity, soil pH level, rainfall, and crop label, which is a categorical variable (i.e., the type of crop). The dataset contained 2200 records, and Table 3 depicts the statistical summary of our dataset for better understanding. Further, in order to understand the true nature of our data before dealing with predictive data analysis, we performed an exploratory data analysis. In this regard, in order to understand what kind of data we were dealing with (range and distribution), we analyzed the distribution of features as depicted in Figure 6. Next, we plotted a correlation heatmap depicting the correlation matrix representing the correlation between different features on the dataset, as shown in Figure 7. According to the correlation heatmap, it is evident that there is only a strong positive correlation (correlation score close to 1) between a few of the features, with most of the features having a very weak correlation or being negatively correlated (correlation score close to 0 or less than 0).

According to the features of the dataset, the soil N, P, and K values hold a key place, from a biological perspective, as they act as the key macro-nutrients that plants feed on while they are growing. In general, the main contributions of these macro-nutrients can be categorized as follows:N—Nitrogen is mostly responsible for the growth of leaves on the plant.P—Phosphorus is mostly responsible for development of flowers, fruits, and growth of roots.K—Potassium is responsible for being able to perform the overall functions of the plant correctly.

These macro-nutrients can be supplied through fertilizer, and depending on the N, P, and K concentrations of the fertilizer, it will be better suited to different ranges of crops. Furthermore, crops require large amounts of N, P, and K to grow and thrive, and plants that are well fed are healthier and more productive. However, if farmers do not use fertilizer, the soil may not provide enough nutrients for maximum growth [4]. Fertilizer adds nutrients that the soil lacks, and understanding the NPK ratios that crops require would thus assist farmers in achieving optimal plant development and yield by managing fertilization. Hence, in this regard, and as depicted in Figure 8, we evaluated the N, P, and K requirements for different types of crops in our dataset. Based on Figure 8, it is evident that apples and grapes require a high potassium level compared to all other crops, based on our data in the dataset.

The rest of the features in the dataset include air temperature, air humidity, soil pH, and rainfall. These features also aid in determining which crop to harvest, as soil pH influences the availability of essential nutrients, bacteria, and toxic elements in the soil; rainfall is essential for the survival of plants; the water requirements of plants may depend on the type of the plants; and air temperature is essential for photosynthesis—when the temperature rises, photosynthesis may also increase, and air humidity is essential for plant transpiration, for example, when the humidity level is high or there is a lack of air circulation, a plant cannot make water evaporate or draw nutrients from the soil, which would eventually result in rotting of the plant. Taken together, all of these features play a vital role in determining which crop to harvest. Further, when it comes to the crop type that can be predicted based on the other available features, there were several categories of crops, including rice, apple, chickpea, black gram, muskmelon, banana, pomegranate, kidney beans, grapes, cotton, coffee, coconut, mango, papaya, orange, lentil, pigeon peas, and moth beans.

### 4.2. Predictive Data Analysis

The Python programming language wasused to create our predictive ML models, and in the dataset preparation stage, once the dataset was acquired, first, we imported the necessary libraries from Python to perform the data preprocessing, such as NumPy for performing mathematical operations, Matplotlib for plotting the charts, Pandas for dataset manipulation and the Scikit-learn library for predictive data analysis. Secondly, as the dataset may contain some missing data that would hinder the performance of our ML models, we searched for the missing values, which were handled successfully.

In the feature selection phase, we manually selected all features from the dataset for N, P, K, air temperature, air humidity, soil pH level, and rainfall with the aim of choosing the best crop to plant as all the features contribute equally to the growth of the crop in a biological perspective. Then, to perform predictive data analysis, we chose N, P, K air temperature, air humidity, soil pH level, and rainfall as our independent variables and crop label as our dependent variable, which was the name of the crop type.

After choosing the dependent and independent variables, we split out the main dataset into training and testing data sets, with a ratio of 70:30, to perform the predictive analysis. Afterwards, five machine learning models (KNN, DT, RF, XGBoost and SVM) were adopted to perform the predictive data analysis. Before training the models, we finalized the data pre-processing stage, and during the splitting of the dataset into training and test datasets, we randomly split the dataset with a training and test set ratio of 70 to 30 (70:30), as described above. After splitting the dataset, we trained our ML models and we evaluated Accuracy, Precision, Recall, and K-Fold cross-validation scores, which are based on four types True Positive (TP), True Negative (TN), False Positive (FN), and False Negative (FN) for all of the underlying ML algorithms we adopted. In terms of TP, TP is defined as cases that are predicted to be positive, and which are actually positive. TN is defined as cases that are predicted to be negative, and which are actually negative, while FP is when the cases are predicted to be positive, but are actually negative. FN is when the cases are predicted to be negative, and they are actually positive. For the five ML algorithms, we adopted the following four key performance metrics for use to determine the classification performance, including the K-Fold cross-validation score. Equation (1) is used to determine the accuracy, which is based on the accurate and total samples. In general, an accuracy score suggests if a model is being trained properly and how it will perform in general, but it does not provide comprehensive information about how it will be applied to the underlying ML problem.
Accuracy = (TP + TN)/(TP + FP + FN + TN)(1)

Equation (2) measures the precision score, which measures the differential rate of the classifier and presents the proportion of accurately predicted positive observations to all expected positive observations.
Precision = TP/(TP + FP)(2)

Equation (3) measures the recall score which measures the ratio of TP over the total number of true. In simple terms it measures the accurately predicted positive observations for all observations in the actual class.
Recall = TP/(TP + FN)(3)

F1 score is an overall accuracy metric that combines precision and recall. A solid F1 score suggests that there are few FPs and few FNs, and that you are on the right track of recognizing serious threats while avoiding false alarms.
F1 = (2 × Precision × Recall)/(Precision + Recall)(4)

Upon the successful completion of model training, the adopted underlying ML models predicted what type of crop would be more suitable, and in the next subsection, we highlight the predicting performance of ML algorithms we adopted for the design of our recommendation platform using the best predicting algorithm.

### 4.3. Experimental Results

In this section, the experimental results are discussed regarding the predicting performance of ML algorithms we adopted. Table 4 demonstrates the accuracy, precision, recall, F1, and 10-fold cross-validation scores we obtained for better comparison. Further, Figure 9 demonstrates the accuracy comparison of all ML models we have adopted.

According to the results we obtained, it is evident that all ML algorithms adopted have varying predicting capabilities in terms of predicting which crop is more suitable according to the input data. In terms of accuracy, it is evident that RF performs the best, with a score of 97.18%, as opposed to other predicting algorithms KNN (96.36%), DT (86.64%), SVM (87.38%), and XGBoost (95.62%). According to the precision score, which measures the proportion of positively predicted labels that are actually correct, both RF and KNN perform equally well, with a score of 97%, whereas DT performs the worst, with a score of 82%. In terms of Recall score, which is about our ML model’s ability to correctly predict the positives out of actual positives, yet again, RF performs with a score of 97%, whereas DT and SVM both perform the worst, with a score of 87%. Next, as per the F1 score, RF is highest, with a rate of 97%, whereas KNN is 96%, DT is 83%, XGBoost is 96% and SVM is 87%.

To evaluate the generalizing capacity of the adopted ML models, we used K-Fold cross-validation with the intention of estimating the overall performance of the models with K = 10. In terms of K-Fold scores, it is clear that RF possesses the highest score of 97.40%. On the other hand, DT performs poorly, with a low score of 83%. Even though accuracy is not a good score to measure the performance of an underlying ML model, based on the other performance metrics such as precision, recall, F1 score, and 10-fold cross-validation scores it is evident that RF, outperforms other ML models while predicting which crop to harvest.

### 4.4. Implementation of the Crop Recommendation Platform

Based on the performance evaluation criteria outlined above, it is evident that RF performs best among the ML algorithms adopted, in terms of predicting which crop to harvest. Hence, for our crop recommendation platform, we intend to use RF to predict the most suitable crop, according to the input parameters submitted by the user, which include N, P, K, air temperature, air humidity, soil pH level and rainfall. Once the best-performing model has been selected, the model is separately serialized/saved for the design of the crop recommendation platform using the Python pickle module as the next step. All steps involved in the design and deployment of the crop recommendation platform are depicted in Figure 10.

For the design of the platform, we used Flask, which is a Python-based microframework used for developing websites that allows developers to design Restful Application Programming Interfaces (APIs) using the Python language in a convenient way. In this regard, in order to design the web pages that the users are interacting with, we designed the necessary web pages using HTML (Hypertext Markup Language), in an interactive way using the JavaScript and CSS programming languages. Once the design was ready, we deployed our system using the local Flask web server and tested its functionality, and whether it was accepting user input with appropriate validations and giving the output as expected. Once the local testing ha been performed, we deployed our local tested web app to Google Cloud App Engine as a Platform as a Service (PAAS), in which the computing resources can be upgraded at any time, guaranteeing almost 99.9% uptime and 24 × 7 convenient access from any device. As per the demonstration purpose, we used their free tier service, which is free of charge within specified monthly usage limits. Figure 11 and Figure 12 showcase the cloud-hosted fully functional recommendation platform.

### 4.5. Discussion

On the basis of the performance metrics we obtained during the training phase of the ML models, it is evident that RF outperforms all of the other ML models we trained in terms of all of the performance criteria we analyzed. Even though the dataset we used for training contained 2200 entries, which is quite a small amount for training, in terms of other performance criteria we evaluated (recall, precision etc.) apart from accuracy, it is evident that RF performs better than the other models. Thus, we used RF as the underlying ML model in our crop recommendation platform. Once the user has submitted the necessary parameters, the system validates whether there are null values and whether the values are within the validation ranges, before submitting the user’s input to the system for further processing, where the system itself processes the inputs and predicts the most suitable crop for planting on the basis of the input parameters, which is highly beneficial for farmers before harvesting, allowing them to make precise decisions about what to harvest, thus resulting in a higher return on investment with less loss. While designing our recommendation platform, we assumed that farmers would obtain this information by submitting parameters using the meteorological data already available and from the widely available existing IoT precision farming solutions, some of which are highlighted in Table 4. As the platform we designed is a cloud-based platform, anyone can access the platform from anywhere at any time with any device, making the service highly convenient. On the other hand, because the dataset we picked to develop our platform was based on data from India, more data may be required from different geographic locations around the world in order to offer our solution to everyone in the world, as crop growth varies from that in India depending on the climate and the environment in other countries. However, as the platform evolves, we may be able to aggregate more data from various regions around the world, in which case users/farmers will be able to first select their geographic location and enter the necessary parameters, after which the system will predict which crop would be most suitable. As our study demonstrates the involvement of ML in the design of precision farming solutions, this could pave the way for future researchers to design real-time prediction systems in combination with IoT.

In general, with agriculture being such an important element of every economy, it is critical to guarantee that even the smallest investment made in the agriculture sector are taken care of, and choosing the best crop for harvesting is a key investment that guarantees a higher-quality harvest. As a result, it is critical to verify that the correct crop has been chosen for the land and according to the environmental context. With our proposed solution, after applying some feature enhancements, such as with the incorporation of an inbuilt IoT hardware set-up for accurate data gathering according to the specific geographic location, and relying on more accurate data collected from different locations and in combination with more parameters with the aid of IoT, it will be possible to offer this technology to everyone, which would be highly beneficial for every farmer that is keen to move into technology-driven precision farming. For the time being, there are a lot of organizations that are engaged in designing precision farming solutions, and a lot of startup companies are also being established that aim to expand the precision farming market and reach out to more farmers. As we have reviewed, most of the available solutions are offered on a subscription basis and are offered as cloud-based solutions; however, trial versions are also available with limited features, with farmers needing to pay more to access additional features. On the other hand, even though there are a lot of commercial solutions are there, there are a few free and open-source (when software is open source, it grants users the right to use, study, change and distribute the software and its source code to anyone and for any purpose) precision farming solutions available that many peoples are not aware of, each having different benefits. Thus, it is indeed essential that farmers are well aware of these solutions, as they will allow them to use technology for free rather than investing higher upfront costs for everything. For better understanding, the Table 5 showcases several of the best available free and open-source precision farming solutions [43,44,45,46,47,48].

From our perspective, as we have discussed, free and open-source precision farming solutions are still growing in the market, and there is still a long way to go, as they have to compete with commercial solutions. On the other hand, most of the free and open-source solutions are aided by a community of developers and other relevant stakeholders and farmers, thus empowering the growth of open-source solutions, as any development issues and doubts related to the integration of technology can be tackled with ease [49,50,51] with the help of these communities. Therefore, the main work in our study, we proposed and designed our crop recommendations platform while bearing this in mind.

According to the summarized table in Section 3, it can be noted that many researchers have presented AI-powered solutions that are applicable to various aspects of agriculture, such as for crop management and soil management. On the other hand, only a few research works have been carried out in this area in recent years, and most of these works have been focused on providing theoretical overviews and practical implementations. None of these studies addressed or discussed how to offer these technologies for free and open source, or how to reach out to a larger audience (farmers) with greater visibility. In contrast, in our study, we demonstrated solutions to these problems, with a step-by-step explanation, and provided an overview of how to offer these technologies for free by giving examples.

## 5. Conclusions

Agriculture, being the primary industry in the world, aids in feeding billions of people all globally. With the involvement of technology, traditional agriculture has transformed such that there is less manual labor, while still achieving better yield and a higher-quality harvest. IoT-enabled smart sensors, underlying communication technologies, actuators, satesatellite, UAV solutions, along with AI, are some of the major technological innovations leveraged in the field of agriculture to reach the next level. This collation of fruitful technologies makes it possible to gather real-time data and make timely and precise decisions without the need for human support, making farming more efficient. AI is the key founding technology of precision agriculture, and resolves complex solutions without human intervention, assisting farmers in making precise decisions regarding the underlying condition of their farms in a timely manner. Currently, most countries are moving towards the adoption of precision farming practices, to take advantage of their immense benefits, such as access to remote monitoring even during the time of the COVID-19 global pandemic, reduced manual labor, and higher harvest. Thus, in this study, we demonstrated a novel cloud-enabled ML-driven crop recommendation platform with a detailed explanation of its step-by-step implementation. Nevertheless, we further provided a brief overview of precision farming, as well as the use of AI in precision farming, summarizing the most recent work carried out in this subject area.

At the present time, the precision farming market is expanding at a rapid rate, and a variety of applications are already on the market. Most of the solutions on the market are commercialized, although there is still free and open-source software available, which many are not aware of. In this regard, we also provided a brief overview of the kinds of solutions already available, and what services they offer. As the key objective of this study is to demonstrate the integration of ML into precision farming, as well as other enabling technologies, like the cloud, we showcased all of the steps involved in designing a cloud-hosted ML-powered crop recommendation platform that can help farmers in deciding which crops to harvest according to the local environmental conditions. Furthermore, we note that these technologies can be freely offered to everyone, and they should be backed by support from communities who are like-minded groups of people interested in these technologies and who have a mind to make this world a better place for everyone. By doing so, we believe that these precision farming solutions and services can reach many farmers, even farmers located in rural and remote areas, ultimately resulting in the growth of precision farming, and allowing most of the challenges associated with feeding billions of people all around the world to be overcome.

## Figures and Tables

**Figure 1 sensors-22-06299-f001:**
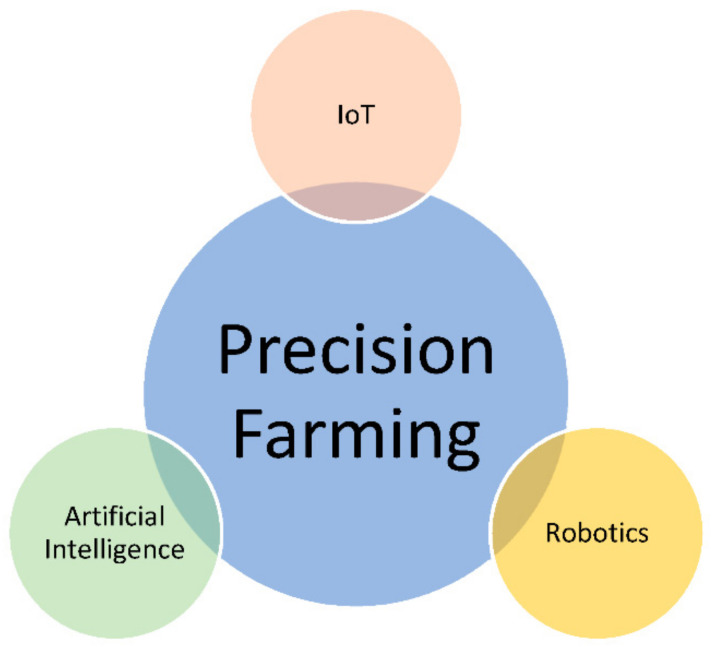
Three foundation technologies of precision farming.

**Figure 2 sensors-22-06299-f002:**
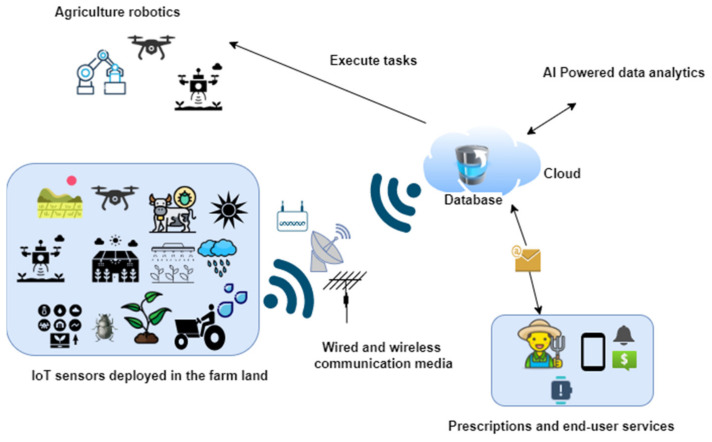
Overall steps in precision farming.

**Figure 3 sensors-22-06299-f003:**
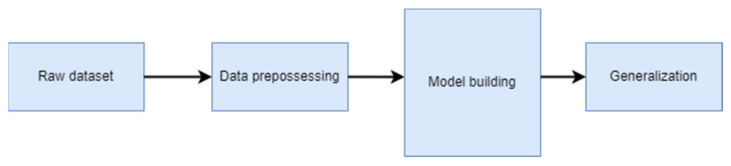
A typical machine learning process.

**Figure 4 sensors-22-06299-f004:**
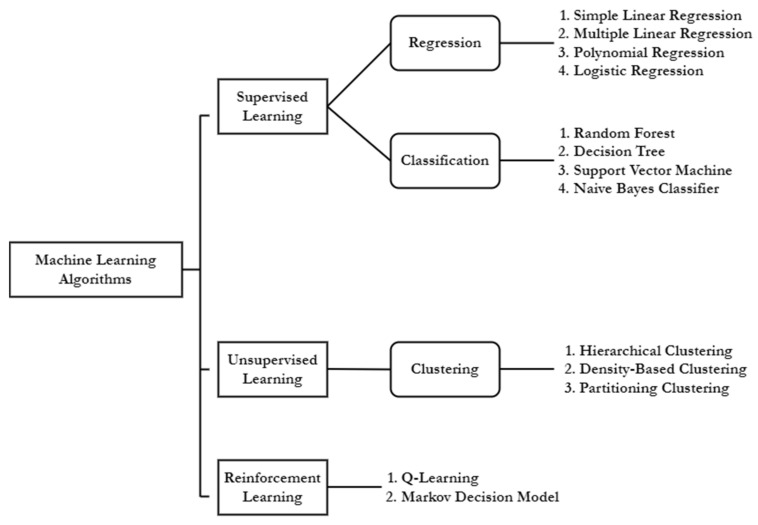
Categories of ML algorithms with key examples.

**Figure 5 sensors-22-06299-f005:**
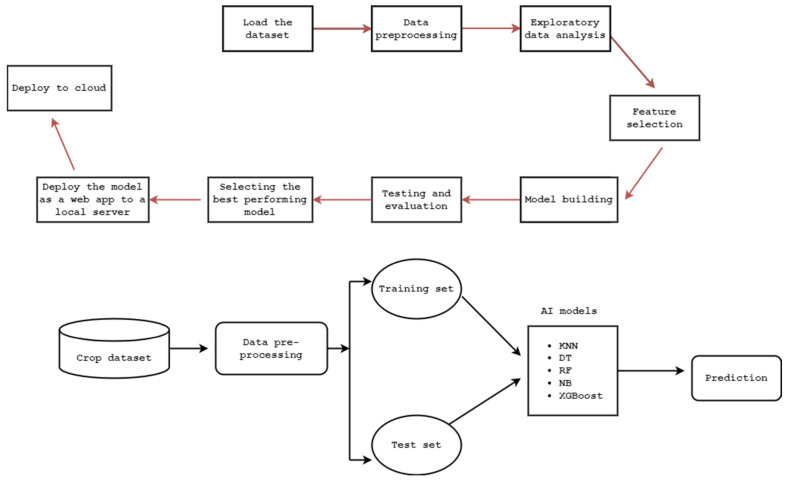
Steps involved in the design of the crop recommendation platform.

**Figure 6 sensors-22-06299-f006:**
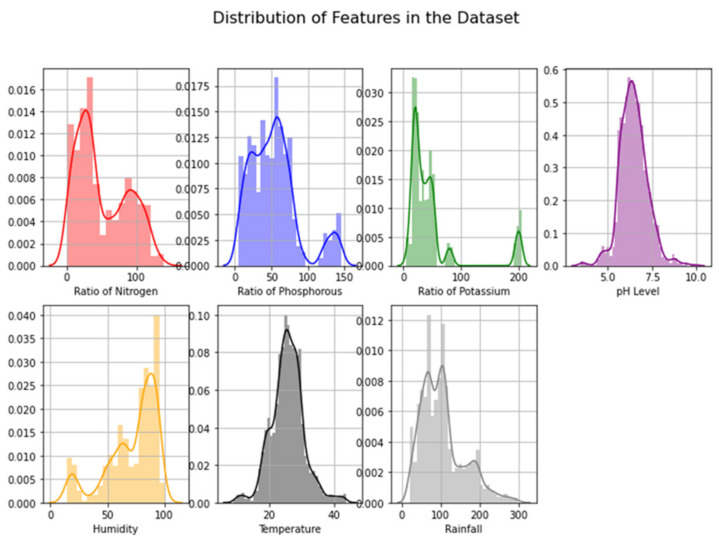
Distribution of features in the dataset.

**Figure 7 sensors-22-06299-f007:**
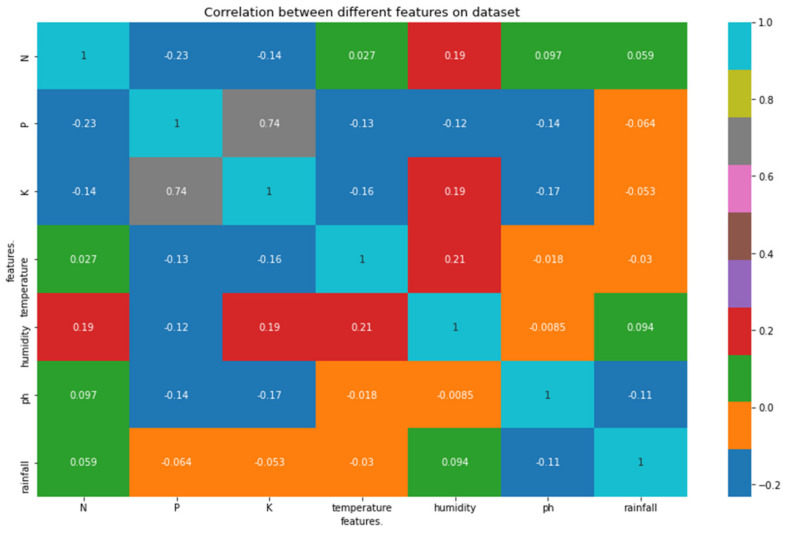
Correlation matrix showcasing correlation between different features of the dataset.

**Figure 8 sensors-22-06299-f008:**
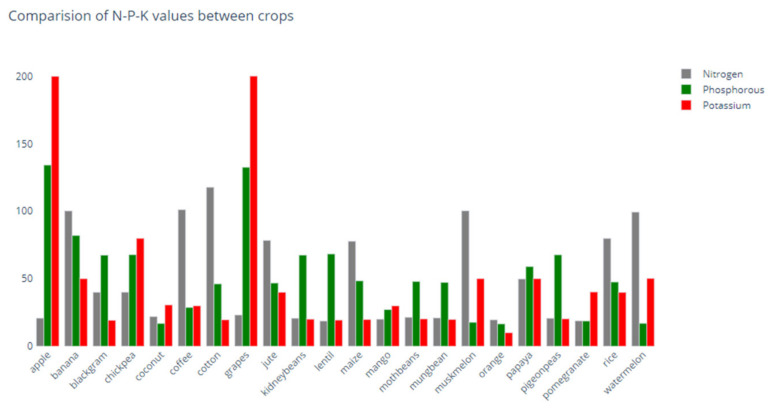
Comparison of N, P, K requirements of different crops.

**Figure 9 sensors-22-06299-f009:**
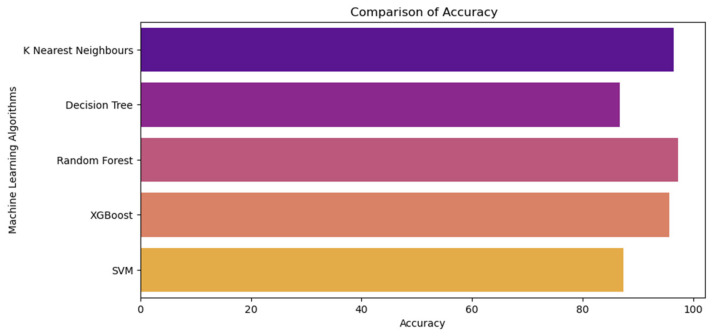
Accuracy comparison of ML algorithms.

**Figure 10 sensors-22-06299-f010:**
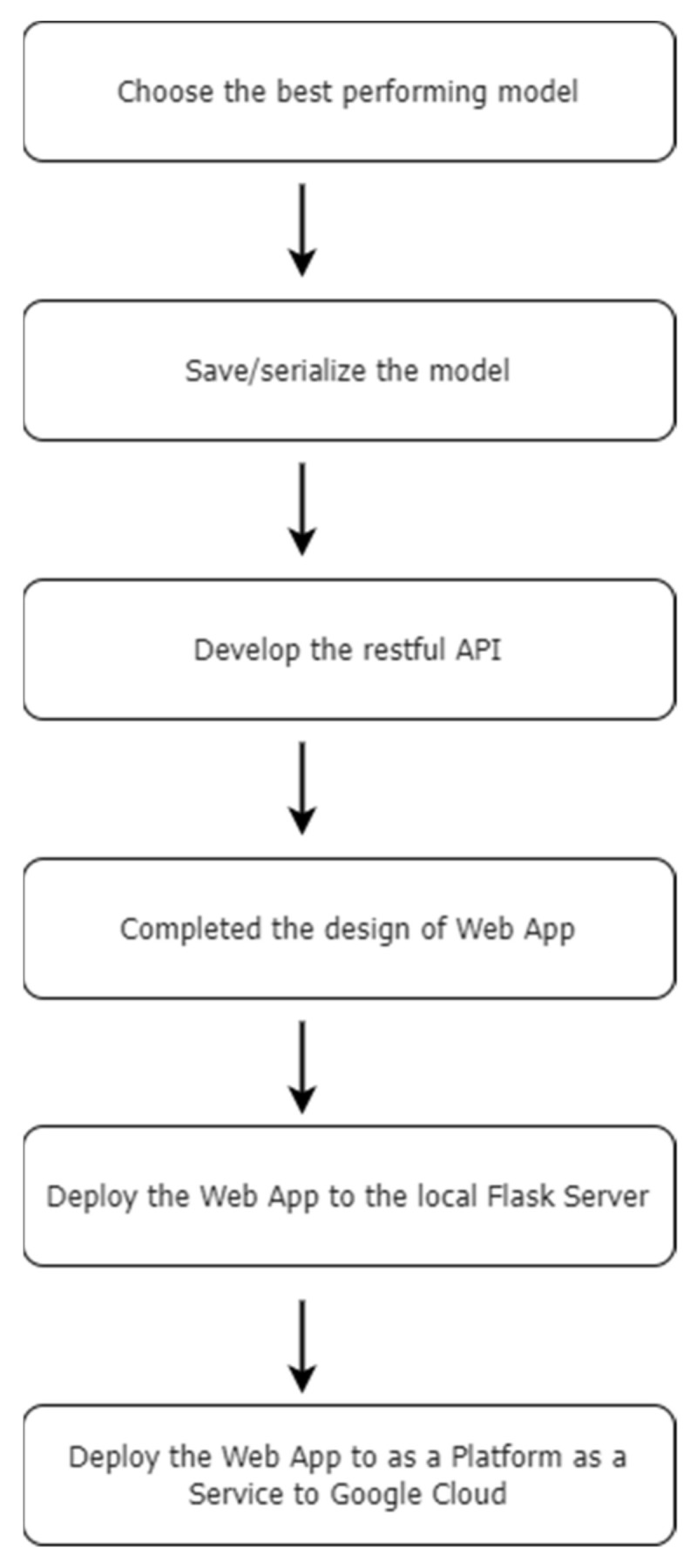
Steps involved in design and deployment of the crop recommendation platform.

**Figure 11 sensors-22-06299-f011:**
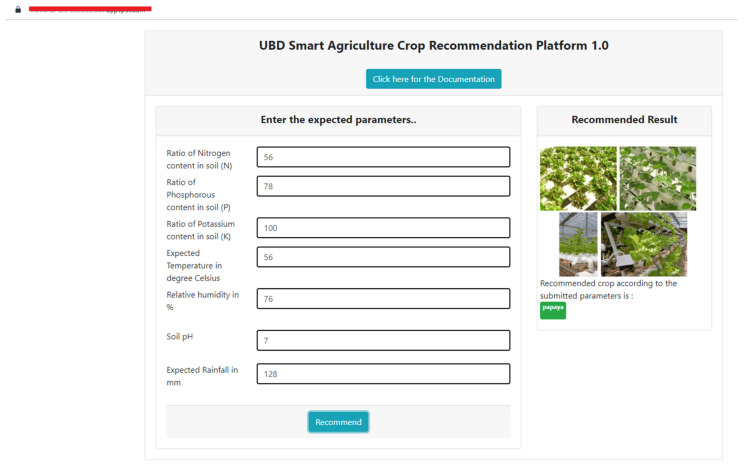
Cloud-hosted crop recommendation platform.

**Figure 12 sensors-22-06299-f012:**
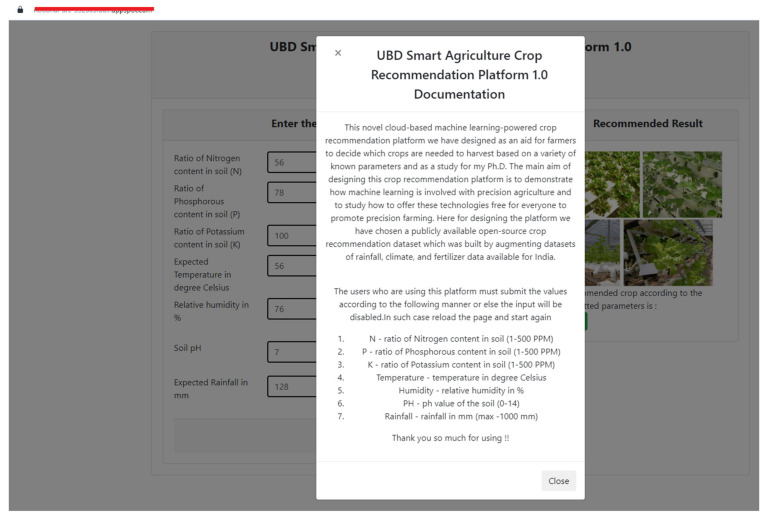
Cloud-hosted crop recommendation platform user documentation.

**Table 1 sensors-22-06299-t001:** Machine learning algorithms.

Machine Learning Algorithm	Description
Regression algorithms	Regression algorithms are supervised learning algorithms that use training data to predict the output numerical value of unknown input. Some of the most frequent regression ML methods include simple and linear regression, polynomial regression, and logistic regression [1,3,7,12].
Random Forest (RF)	RF is an ensemble learning methodology for classification and regression that works by constructing a jumble of decision trees at training time and outputting the category that is the mode of the categories or mean prediction of the individual trees [7,9,12].
Decision Tree (DT)	DT is a classification and regression algorithm that works with both categorical and continuous input and output variables. It divides the data into two or more homogeneous sets or areas based on the independent variables’ most significant splitter [1,2,3,4,7,12].
Support Vector Machine (SVM)	SVM is a classification and regression algorithm that creates multi-dimensional planes (boundaries) between data points in the feature space [7]. The SVM predicts the output based on the training data separated into classes and SVM is better suited to high-dimensional data with many predictor variables [1,2,3,12].
K-Nearest Neighbors (KNN)	KNN is a supervised learning algorithm that divides a labeled dataset into classes depending on its outputs. As a result, a new forthcoming item is given a class depending on its K nearest neighbors [3,4,5,6,7,9,12].
Naive Bayes (NB)	A Naive Bayes classifier is a probabilistic machine learning model for classification problems. The Bayes theorem is the foundation of the classifier [1,2,3,4,7,12].
Extreme Gradient Boosting (XGBoost)	XGBoost is a regression and classification algorithm built on the principles of gradient boosting framework [1,2,3,4,5,6,7,9,12].
Clustering algorithms	In contrast to supervised learning, clustering algorithms automatically uncover natural grouping in data and can only interpret the input data and locate natural groups or clusters in feature space [7,9,12].

**Table 2 sensors-22-06299-t002:** Summary of the most recent studies with their contributions.

Reference	Application Area/s	ML Algorithm/s Used	Research Contributions
Kumar et al. (2019)	Crop management	SVM, DT, Logistic Regression (LR)	The authors introduced an ML-powered recommendation system for identifying crop suitability and pest control. Further, they found that among the ML algorithms they used, SVM gave the best results as opposed to other algorithms.
Shinde et al. (2015)	Crop management	RF	The study proposed using data mining techniques to provide recommendations for what crops to grow, crop rotation, and fertilizer identification in the form of web-based and smart mobile applications.
Mahir et al. (2008)	Soil management	Neural Network (NN)	Considering the various parameters of the soil the authors presented a recommendation system for recommending what kind of crop to harvest in certain types of soil.
Arooj et al. (2018)	Soil management	NB, DT, NN, SVM	The authors provided an empirical study on various data mining classification algorithms to classify the datasets of different regions based on the soil properties.
Rajak et al. (2017)	Crop management and soil management	SVM, Artificial Neural Network (ANN)	The authors made a recommendation system to recommend crops for farming sites based on the type of soil, where they used data from a soil sampling lab to train their underlying ML models.
Pudumalar et al. (2017)	Crop management	KNN, NB, RF	The authors proposed an ensemble model with a majority voting technique to recommend crops according to site-specific parameters.
Alam et al. (2020)	Crop management	RF	The authors presented a real-time computer vision-based crop/weed detection system for variable-rate agrochemical spraying.
Brunelli et al. (2019)	Crop management	NN	The authors presented a near-sensor neural network algorithm that can automatically detect the Codling Moth in apple orchids. Once the insect is detected, the system itself performs classification and sends a real-time alert to the farmers.
Tsouros et al. (2019)	Crop management	-	The authors summarized the data acquisition methods and technologies for acquiring images in UAV-based precision farming in their study and provided a comparison.
Treboux and Genoud. (2019)	Crop management	DT, RF	The authors presented the performance of ML algorithms used in aerial image detection in precision farming.
Dimitriadis and Goumopoulos. (2008)	Water management	NB, ZeroR, OneR, J48, DecisionStump	ML techniques were used by the authors to automatically extract new knowledge in the form of generalized decision rules for the optimum management of natural resources such as water in farming land.
Reddy et al. (2020)	Water management	DT	The authors proposed a real-time smart irrigation system powered by the DT ML algorithm to alert ranchers in real time about when to supply water to the field.
Junior et al. (2022)	Crop management	K-Means Clustering	To reduce the data congestion when overloading IoT data to the cloud, the authors proposed an approach for collecting and storing data in a fog-based smart agriculture environment with different data reduction techniques.
Shukla et al. (2021)	Crop management	LR, KNN, SVM, RF, NN	The authors introduced an IoT and ML-powered platform capable of monitoring the condition of crops and crop disease detection. Moreover, the introduced system was also linked with UAV, and through the multispectral images captured through IoT integrated with UAV, the system was able to detect the health of crops in the field.
Petropoulos et al. (2020)	Crop management	Support Vector Regression (SVR), Random Forest Regression (RFR)	The authors proposed novel ML and DL techniques to predict yield and plant growth variation in controlled greenhouse environments. In this regard the authors have deployed Recurrent Neural Network (RNN), using the Long Short-Term Memory (LSTM) neuron model for the prediction and they have presented a comparative study using SVR and RFR ML models.
Agarwal and Tarar, (2021)	Crop management, Soil management	SVM	The authors provided a novel AI model for predicting the type of the crop to harvest based on the characteristics of the soil and in that regard, they have used SVM as the ML model and RNN and LSTM as DL algorithms.
Raja et al. (2018)	Crop management	Regression algorithms	The authors performed an experiment with past data to predict the crop yield and price that a farmer can obtain from his land using regression classification techniques.
Viviliya and Vaidhehi. (2019)	Crop management	NB, J48, Association rule learning	The authors proposed a hybrid crop recommendation system for recommending crops to South Indian states by considering various environmental attributes.
Goap et al. (2018)	Water management	K-Means clustering, SVR	The authors presented a novel open-source technology-based smart irrigation system that predicts irrigation requirements for fields using a variety of environmental parameters, in which they have came up with a novel algorithm for this purpose.
Brock et al. (2018)	Livestock management	Self-organizing maps	The authors presented a new approach for classifying herd types in livestock systems by combining expert knowledge and a machine-learning algorithm known as self-organizing maps (SOMs), which they applied practically to the cattle sector in Ireland, in order to understand ongoing discussions surrounding control and surveillance for endemic cattle diseases.
Lee, M. (2018)	Livestock management	RF, Expectation maximization	This study proposed and implemented a system to analyze 3-axis acceleration data from IoT sensors and compared the pattern-recognition performance of machine-learning algorithms for three breeding cow behavioral patterns: estrus start, peak estrus activities, and estrus finish.

**Table 3 sensors-22-06299-t003:** Statistical summary of our dataset.

Statistics	N	P	K	Air Temperature	Air Humidity	Soil pH	Rainfall
Entries	2200	2200	2200	2200	2200	2200	2200
Mean	50.55	53.36	48.14	25.61	71.48	6.47	103.50
Standard Deviation	36.91	32.98	50.64	5.06	22.26	0.77	54.95
Minimum	0.00	5.00	5.00	8.82	14.25	3.50	20.21
Maximum	140.00	145.00	205.00	43.67	99.98	9.93	298.56

**Table 4 sensors-22-06299-t004:** Accuracy, precision, recall, F1 and 10-fold cross validation scores.

Model	Accuracy Score	Precision Score	Recall Score	F1 Score	K-Fold Cross Validation Score (K-10)
KNN	96.36%	97%	96%	96%	97%
DT	86.64%	82%	87%	83%	92%
RF	97.18%	97%	97%	97%	97.40%
XGBoost	95.62%	96%	96%	96%	96.31%
SVM	87.38%	87%	87%	87%	88.50%

**Table 5 sensors-22-06299-t005:** Free and open-source precision farming solutions.

Software	Key Features	Development Mode	Web Link
AgroSense [44]	Soil health tracking, overall planning, and budgeting	OpenAPI (this makes the application programming interface publicly available to software developers)	https://agrosense.eu (accessed on 5 June 2022).
Tania [45]	Planning and budgeting, labor planning	OpenAPI	https://usetania.org (accessed on 5 June 2022).
farmOS [46]	Crop management, labor management, order management	OpenAPI	https://farmos.org (accessed on 5 June 2022).
LiteFarm [47]	Crop management	OpenAPI	https://www.litefarm.org (accessed on 5 June 2022).
Granular Insights (even though this is free, it is not an open-source solution) [48]	Crop management, labor management, order processing	Cloud hosted	https://granular.ag/granular-insights/ (accessed on 5 June 2022).

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
