# Peer review of "A Cloud Enabled Crop Recommendation Platform for Machine Learning-Driven Precision Farming"

_sensors, 2022, doi:10.3390/s22166299_

Round 1

Reviewer 1 Report

In this paper, the authors provide an overview of AI-driven precision farming /agriculture with related work and propose a cloud-based ML crop recommendation platform.

The paper is well written and easy to understand. Nonetheless, there are some aspects that should be improved:

- The authors provide a good introduction to the current status and  problems of agriculture

- Line 54 is not easy to understand. Should be rewritten

- Lines 63, 64: too much "ands". Could be improved.

- line 68: "and may even lead" would be better.

- line 70, 71: "which is clearly proved"

- line 84-87 is not clear, should be rewritten to make it easy to read

- line 97: missing comma after AI: "AI, can reshape..."

- line 120: the "a" after the period should be capitalized: "A brief overview"...

- line 123: add a comma after "from others"

- The first two topics of the contributions of the study should be rewritten to be more easy to read and understand

- line 162: missing a "are": Additionally, these 'are' analyzed and ...

- lines 258-264: You should say that DL is a branch of ML. As it is  written, it seems that DL is not ML.

- all tables are too wide. Should have the width of the text

- line 322: you have an extra period: "plant development and agricultural..."

- line 378: how was the data augmented? With what data? With data for  India[22]? If so, it should more clear. Also, what is the size of the data set? That is relevant information that should be included.

- line 395-298: the start of this sentence is no clear.

- line 435: what is the source of the needs for each crop?

- line 447: extra space before the ";"

- Section 4: This section should have more subsections. The feature section should be a section. Starting on line 459, there should be   another section, where you start talking about the tools used. This   section should be better structured.

- The size of the dataset is only briefly mentioned in line 613. This  should be stated earlier.

- Since the data set was based on data from India, it should be discussed how the system will behave in other climates other than   India, and understand if the results are comparable.

Author Response

Dear Editor / Reviewer,

Thank you very much for giving us the opportunity to revise the manuscript. We would like to thank the editor and all the reviewers for their valuable comments and suggestions, which helped a lot for us to improve our work. Accordingly, based on the feedback, we have revised our manuscript. For better understating, in the manuscript we have highlighted the content in RED color (Track Changes), For clarity, we have marked our responses in blue in the attached letter. Again, thank you very much for your valuable comments.

Reviewer 2 Report

The main content of research presented in the paper is a novel cloud-based Machine Learning-powered crop recommendation platform that aid farmers to decide which crops are needed to harvest based on a variety of known parameters.

The topic is not unique, but it is worthy of researching.

The main proposal is a novel cloud enabled Machine Learning-driven crop recommendation platform with a detailed explanation of its step-by-step implementation.

The deduced conclusions based on the research methods are that the precision farming solutions and services can reach out to many farmers, even the farmers who are located in rural and remote areas, ultimately resulting in the growth of precision farming and overcoming most of the challenges associated with feeding billions of people all around the world.

The conclusions are tenable. In my opinion, the article is not sufficiently clear what progress has been made compared with the current research results.

The abstract is informative. It reflects the body of the paper.

The introduction provides sufficient background information for readers in the immediate field to understand the problem.

The text is well arranged and the logic is clear. There are practically no grammatical errors in the article. The related concepts are introduced clearly. The readability is sufficient.

The approaches and techniques used in the study are not new. The novelty lies in its application to a specific situation.

Derivation of formulas is sufficient.

The theoretical analysis is enough for the purposes of the article.

All figures and tables are clear enough to summarize the results for presentation to the readers. In general, all figures and tables are well referred to in the text. However, there are 2 tables with the number 4.

The reference section is informative. Not all references are accurate. Authors should review and correct the formatting of references in the text and in the References section to make them more complete, more homogeneous and in accordance with the journal's rules.

Author Response

(The authors gave the same response as above.)
